# Stochastic treatment regimes in climate-health research: Reassessing malaria risk under warming scenarios in Colombia

Juan David Gutiérrez 🄳 *

Facultad de Ciencias Médicas y de la Salud, Instituto Masira, Universidad de Santander, Bucaramanga, Santander, Colombia

* jdgutierrez@udes.edu.co

## Abstract

Malaria remains a significant public health challenge, mainly because climate change is altering transmission dynamics. This study investigates the relationship between rising temperatures and malaria cases across the 100 municipalities in Colombia with the highest incidence from 2007 to 2023. We employed causal machine learning techniques to analyze how incremental temperature impacts malaria incidence while controlling for valid confounding variables. Our findings reveal that with the currently observed temperature, malaria transmission intensifies with temperatures between 15 and approximately 23.5 °C but declines at higher temperatures, indicating an optimal range for transmission. Our results suggest an exposure-response relationship where higher temperature increases are associated with greater reductions in the probability of excess malaria cases. The Average Treatment Effect (ATE) on excess malaria cases for the evaluated temperature regimes showed a progressive decrease, from -0.007 when temperature increased by 0.5°C to -0.063 when temperatures were increased by 2.0°C, relative to current temperatures. These results suggest that further warming could constrain malaria transmission intensity in regions already experiencing high temperatures. This research underscores the importance of tailored public health strategies that consider local temperature profiles and socio-economic conditions in malaria control efforts.

### Author summary

This study explores the relationship between climate change and malaria transmission in Colombia, focusing on the impact of rising temperatures on malaria incidence. We analyzed data from the 100 municipalities most affected by malaria between 2007 and 2023. Our results indicate that while warmer temperatures can enhance malaria transmission, there is a threshold beyond which transmission declines, revealing a non-linear relationship. Specifically, we found that the

**Data availability statement:** The code and dataset for replicating the causal machine learning analysis are accessible at: https://github.com/juandavidgutier/temperature_malaria_STR.

**Funding:** The author(s) received no specific funding for this work.

**Competing interests:** The authors have declared that no competing interests exist.

Average Treatment Effect on excess malaria cases decreases if future temperatures rise, suggesting that in areas already experiencing high temperatures, further warming may actually reduce transmission risk. These insights highlight the need for targeted interventions that address local climatic and socio-economic factors to effectively manage malaria risk in vulnerable regions.

## 1. Introduction

Malaria is a life-threatening parasitic disease caused by *Plasmodium* species, primarily transmitted to humans through the bite of female *Anopheles* mosquitoes. The etiology of malaria involves the introduction of *Plasmodium* parasites into the human bloodstream, where they infect liver cells and subsequently red blood cells, leading to cycles of fever, chills, and other systemic symptoms [1]. Globally, malaria remains a significant public health challenge, with the WHO estimating over 240 million cases and 627,000 deaths in 2021, predominantly in sub-Saharan Africa. In the Americas, malaria transmission is concentrated in the Amazon basin, with countries like Brazil, Colombia, and Venezuela bearing the highest burden [2,3]. In Colombia, most of the cases occur in the epidemiological regions of Uraba-Bajo Cauca-Sinu-San Jorge, Pacific, and Amazon [4].

The relationship between malaria incidence and temperature is well-documented, as temperature plays a critical role in the life cycle of the *Plasmodium* parasite and the behavior of its mosquito vector, *Anopheles* [5]. Warmer temperatures accelerate the development of *Plasmodium* parasites within mosquitoes, reducing the extrinsic incubation period and increasing the likelihood of transmission. Additionally, higher temperatures can expand the geographical range of malaria vectors to previously unaffected regions, particularly at higher altitudes [5]. However, excessively high temperatures may negatively affect mosquito survival, thereby reducing transmission in some areas. Studies have shown that the optimal temperature range for malaria transmission is between 20 °C and 30 °C, with deviations from this range impacting the incidence rates [6,7]. Climate change, characterized by rising global temperatures, has further exacerbated malaria risks in regions with favorable climatic conditions, while altering transmission patterns in others [5].

Understanding the complex effects of temperature on malaria incidence requires methods that disentangle causality while addressing the nonlinear dynamics of climate-health relationships. Causal machine learning [8,9] provides a framework for this task by combining counterfactual reasoning with flexible estimation algorithms, enabling researchers to model exposure-response curves while adjusting for time-varying confounders [10]. Unlike traditional regression approaches that assume fixed parametric forms, causal machine learning accommodates heterogeneous treatment effects and complex interaction patterns inherent in ecological systems [11]. Stochastic Treatment Regimes (STR) extend this paradigm by modeling incremental temperature shifts as probabilistic interventions rather than deterministic thresholds [12]. This approach is particularly suited to climate-health research, as it

preserves natural variability in exposure distributions while estimating the effects of relevant warming scenarios (e.g., + 1.5 °C targets under climate accords) [13]. By implementing STR through Targeted Maximum Likelihood Estimation (TMLE) [14], researchers can leverage the double robustness property of TMLE to further strengthen causal claims by providing valid inference even under partial model misspecification [14]. Together, these methods address critical limitations of previous climate-malaria studies that relied on correlational analyses or static temperature thresholds, offering a way to quantify how rising temperatures may reshape the malaria burden.

This research seeks to deploy a causal machine learning approach, particularly exposure-response and STR models, to explore whether future rising temperatures contribute to the occurrence malaria cases within the 100 municipalities in Colombia that reported the highest number of cases between 2007 and 2023. The findings from this analysis are designed to bolster surveillance initiatives and guide focused eradication strategies in the country's most vulnerable areas.

## 2. Methods

### 2.1. Ethics statement

The Bioethics Board of the Universidad de Santander provided ethical clearance for our research (Minute No. 002, February 13, 2023). Our investigation follows the STROBE guidelines, which enhance the reporting of observational studies in epidemiology. The data on malaria cases were provided by the SIVIGILA and corresponded to anonymized data; for this reason, the data shared in the GitHub repository do not contain potentially identifying participant information. The dataset provided in the repository contains only aggregated, non-identifiable data. All personal identifying information has been removed during preprocessing to ensure compliance with data protection regulations and ethical research standards.

### 2.2. Malaria cases

We carried out an ecological study (our observation unit was the municipality) over several years (2007–2023) to assess how increasing temperatures influence malaria occurrences in the 100 Colombian municipalities most affected by the disease. The response variable was malaria cases, and the exposure variable was temperature. Additionally, we included a set of climate, ecological, and socio-economic factors as co-variates in our model (Fig 1).

We obtained daily malaria cases that occurred in each municipality since January 2007 to December 2023, the data were accessed on 12th November 2024, from the National Public Health Surveillance System (SIVIGILA), a data source previously anonymized by the national health authorities of Colombia. We organized these daily figures into monthly totals for each municipality, discarding any records that had discrepancies in either occurrence locality of the case, date, or age (e.g., > 120 years, 77 months, etc.). Furthermore, we excluded data from municipalities situated at elevations exceeding 1,600 meters above sea level, as this is recognized as the altitudinal limit for malaria transmission by the National Health Institute [4]. Afterward, we pinpointed the top 100 municipalities with the highest incidence of malaria cases.

We employed the Standardized Incidence Ratio (SIR) as an indicator to assess the occurrence of malaria in a specific municipality against the rates derived from national data. It's important to note that SIR values are positive and centered around 1; a value exceeding 1 signifies that the malaria incidence in the municipality is higher than what would be expected based on national averages.

To tackle the age-related disparities in malaria incidence, we applied the indirect method for age standardization, which accommodates differences in age demographics and allows for meaningful comparisons among municipalities over time [15]. This standardization utilized age categories established by the WHO [16] and national population projections [17]. We utilized the epitools package (version 0.5-10.1) in R software [18] to calculate the SIR for each municipality on a monthly basis. Additionally, we defined excess malaria cases as a binary variable, assigning a value of 1 when SIR > 1, and 0 for all other SIR values.

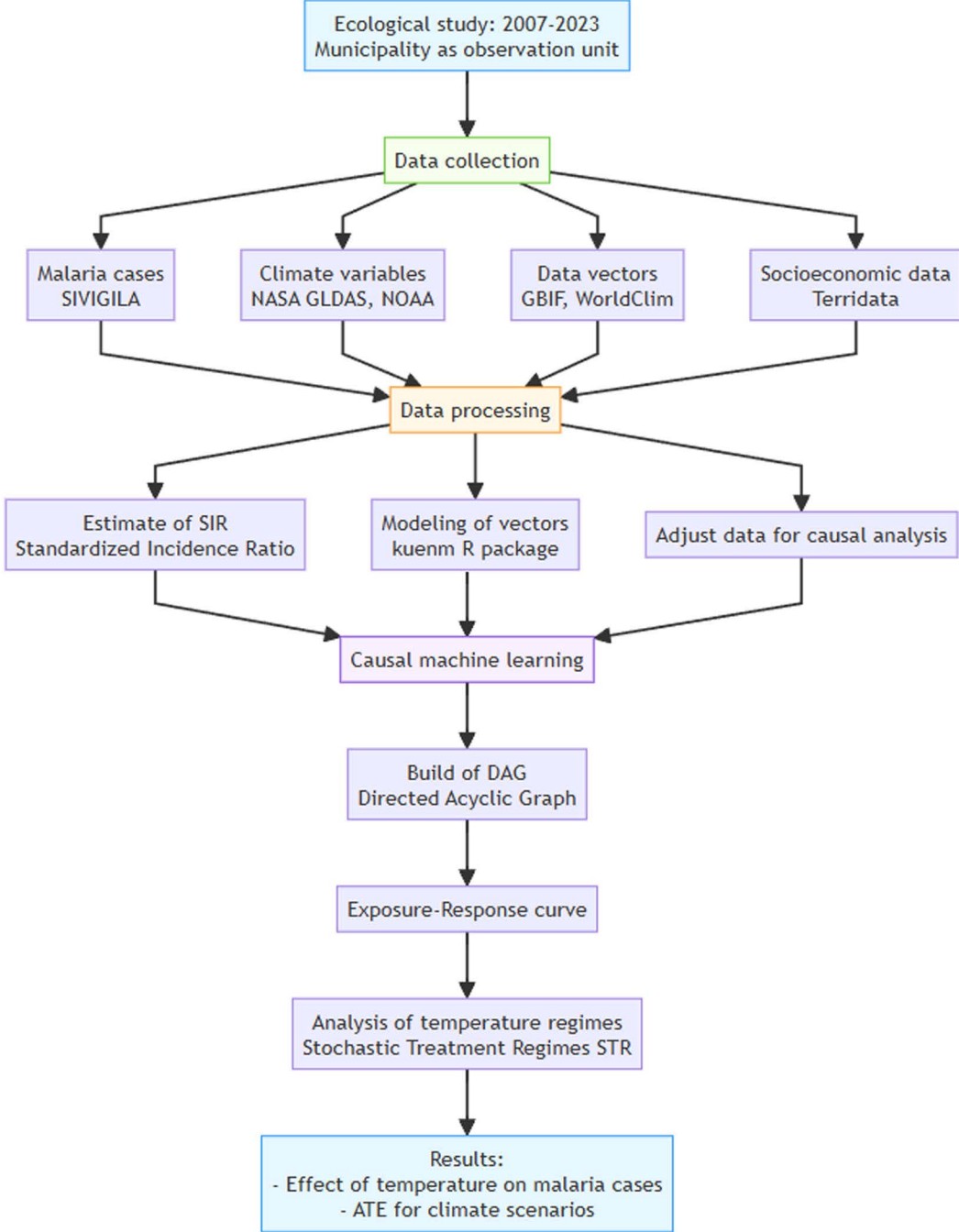

**Fig 1. Flowchart of the analysis steps and methods used in the study (see details in sections 2.1 to 2.5).**

## 2.3. Climate variables

We acquired monthly records of temperature and precipitation spanning from January 2007 to December 2023 through the NASA product GLDAS_NOAH025_M version 2.1 [19], which has a spatial resolution of 0.25 degrees. To ensure proper alignment between raster layers and municipal boundaries, as well as to compute the monthly averages for each variable,

we utilized the raster package in R (version 4.0.3) [20]. Additionally, we sourced monthly data on eleven atmospheric and oceanic indices (Table 1) from the National Oceanic and Atmospheric Administration (NOAA) for the same time frame [21].

## 2.4. Municipality area with malaria vectors' co-occurrence

We focused on eight *Anopheles* species known to transmit malaria in Colombia: *An. albimanus*, *An. darlingi*, *An. nuneztovari*, *An. calderoni*, *An. oswaldoi*, *An. pseudopunctipennis*, *An. punctimacula*, and *An. Rangeli*. Occurrence data for these vectors were extracted from the Global Biodiversity Information Facility [22]. To minimize spatial autocorrelation, we applied spatial filtering using the thin function from R's spThin package [23], employing a 25 km$^2$ thinning parameter with 100 repetitions, which resulted in one occurrence point per cell. This process generated species-specific databases containing filtered geographic coordinates.

To predict potential vector distribution, we implemented the Maximum Entropy algorithm through the kuenm R package [24]. Environmental inputs consisted of all 19 bioclimatic variables obtained from WorldClim version 2.1 [25] at a 30-second spatial resolution (approximately 1 km$^2$ at the equator). These variables represent ecologically meaningful summaries of temperature and precipitation patterns:

Temperature-related variables: Bio1 (Annual Mean Temperature), Bio2 (Mean Diurnal Range), Bio3 (Isothermality), Bio4 (Temperature Seasonality), Bio5 (Max Temperature of Warmest Month), Bio6 (Min Temperature of Coldest Month), Bio7 (Temperature Annual Range), Bio8 (Mean Temperature of Wettest Quarter), Bio9 (Mean Temperature of Driest Quarter), Bio10 (Mean Temperature of Warmest Quarter), and Bio11 (Mean Temperature of Coldest Quarter).

Precipitation-related variables: Bio12 (Annual Precipitation), Bio13 (Precipitation of Wettest Month), Bio14 (Precipitation of Driest Month), Bio15 (Precipitation Seasonality), Bio16 (Precipitation of Wettest Quarter), Bio17 (Precipitation of Driest Quarter), Bio18 (Precipitation of Warmest Quarter), and Bio19 (Precipitation of Coldest Quarter).

Variable selection was conducted through an exhaustive approach as recommended by Cobos et al. (2019) [24]. First, we calculated the correlation matrix among all 19 bioclimatic variables using occurrence points for each vector species. Then, we generated 10 distinct variable subsets by systematically removing highly correlated variables (r ≥ 0.7) while maintaining ecological representation. For each subset, we identified correlated variable groups and randomly retained one representative variable per group to minimize multicollinearity while preserving environmental information diversity. This approach allowed us to evaluate multiple plausible environmental hypotheses rather than relying on a single variable selection.

**Table 1. Monthly atmospheric and oceanic indices downloaded from the National Oceanic and Atmospheric Administration (NOAA).**

| Index | Detail |
|---|---|
| SST12 | Sea surface temperature in El Niño region 1–2 |
| SST3 | Sea surface temperature in El Niño region 3 |
| SST34 | Sea surface temperature in El Niño region 3–4 |
| SST4 | Sea surface temperature in El Niño region 4 |
| NATL | Sea surface temperature North Atlantic (5–20°North, 60–30°West) |
| SATL | Sea surface temperature South Atlantic (0–20°South, 30°West-10°East) |
| TROP | Sea surface temperature Global Tropics (10°South-10°North, 0–360) |
| SOI | Sea level pressure standardized Tahiti - standardized Darwin |
| Equatorial SOI | Sea level pressure standardized anomalies Indonesia |
| CPOLR | Central Pacific outgoing long wave radiation (170°E-140°W,5°S-5°N) |
| Winds Equator | 200 millibar zonal winds (165°West-110°West) |

For model calibration, we tested 16 regularization multiplier values (0.1, 0.2, 0.3, 0.4, 0.5, 0.6, 0.7, 0.8, 0.9, 1.0, 2.0, 3.0, 4.0, 5.0, 6.0, and 10.0) and all 31 possible combinations of feature classes (linear = l, quadratic = q, product = p, threshold = t, and hinge = h). This resulted in 4,960 candidate models per vector (10 variable subsets × 16 regularization multipliers × 31 feature class combinations). Model selection was conducted based on three criteria: (1) statistical significance assessed through partial ROC analysis with 500 iterations and 50% bootstrap resampling (models with p-value < 0.05 were considered significant); (2) omission rates below 5% using test occurrence data; and (3) model parsimony determined by Akaike's information criterion corrected for small sample sizes (AICc), where models with ΔAICc ≤ 2 from the best model were considered equally plausible. The specific model parameters selected for each vector can be obtained by reproducing the code shared on GitHub.

For each vector species, we converted continuous suitability predictions into binary presence-absence maps using the lowest training presence threshold approach with a 5% omission error rate [26]. This threshold represents the minimum suitability value from the logistic output where presence locations were recorded. We then combined these binary distribution maps to quantify potential vector co-occurrence of four or more vector species for each municipality using the raster package in R (version 4.0.3) [20].

## 2.5. Socio-economic data

We obtained from the Terridata repository [27] the percentage of households with multidimensional poverty, and incorporated this data for each municipality. The multidimensional poverty serves as a comprehensive measure of social and economic vulnerability, encompassing factors such as access to healthcare, educational attainment, and living conditions. The percentage of households with multidimensional poverty in each municipality has a unique measure in 2018 because this variable was measured by the National Department of Statistics, in the last national population census.

## 2.6. Causal machine learning implementation

### 2.6.1. Directed Acyclic Graph (DAG).
We constructed a DAG [28] with malaria cases as the outcome variable and temperature as the exposure variable (Fig 2). Given that hydro-climatic patterns in continental regions are influenced by atmospheric and oceanic conditions [29–32], we considered the eleven atmospheric and oceanic indices as potential confounders of the effect of temperature on malaria cases. Atmospheric and oceanic indices act as confounding factors because they can influence monthly temperature while simultaneously affecting other hydroclimatic variables, such as rainfall and humidity, which may also impact malaria incidence [33–37].

Year and month were included as potential confounders because both variables define the annual trend and seasonality of the exposure (temperature) and the outcome (malaria cases). Rainfall was considered as another potential confounding factor because this variable can be simultaneously associated with the exposure and the outcome variables [38].

The percentage of the area within each municipality suitable for supporting four or more vector species, and the percentage of households with multidimensional poverty, were also incorporated into our model. Note that in the DAG the nodes representing temperature, rainfall, and vectors (percentage of the area within each municipality suitable for supporting four or more vector species) are linked to the node multidimensional poverty, because since the Spanish colonizers first arrived in Colombia, areas with extreme weather conditions—such as intense heat and heavy precipitation— and malaria presence, have been identified as having unhealthy climates. As a result, successive governments have systematically overlooked in terms of social advancement and the provision of essential services for these communities inhabited mainly by Indigenous and Afro-descendant populations [39,40].

We used the R package DAGitty (version 0.3-4) [41] to test the conditional independence assumptions (i.e., to identify whether two variables are related when controlling for a third variable) about the causal structure represented by the DAG. The appropriate adjustment to the causal model was identified using the R package ggdag (version 0.2.12) [42].

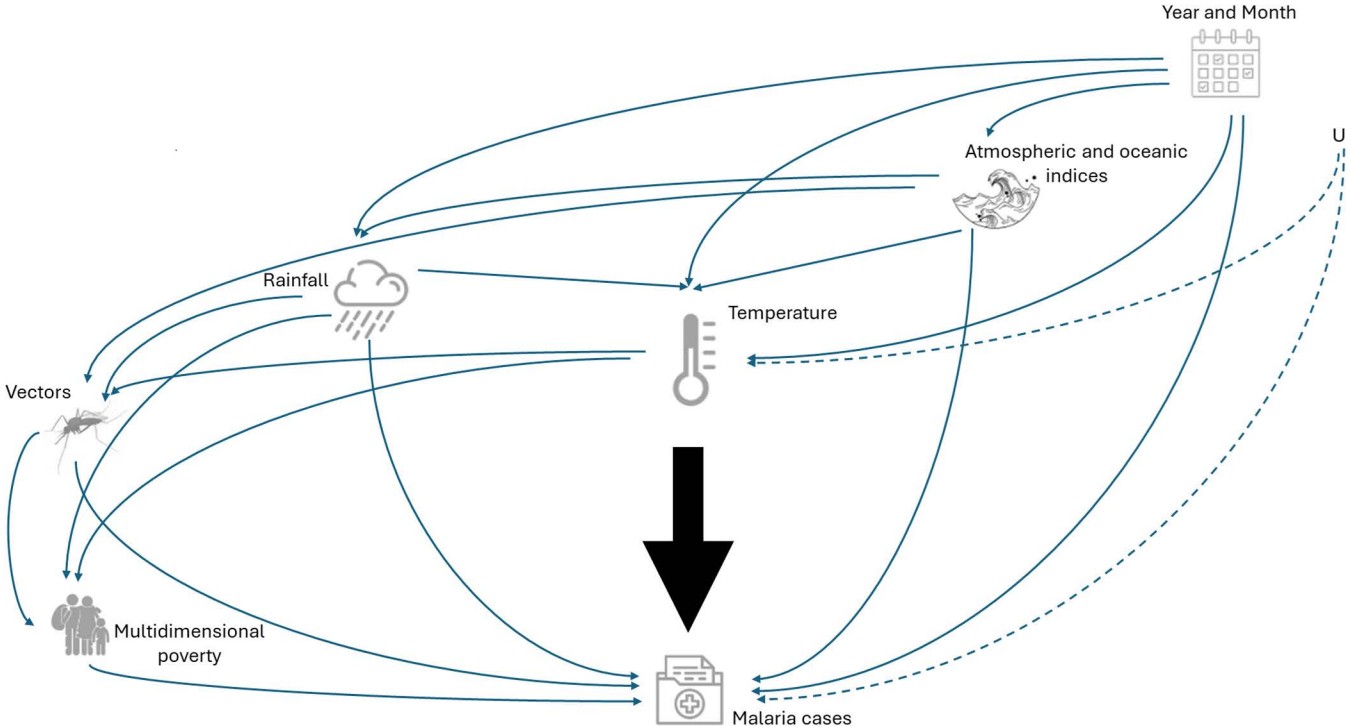

**Fig 2. Directed Acyclic Graph (DAG) illustrating the proposed causal link between temperature (exposure variable) and malaria cases (outcome variable).** The black arrow indicates the causal relationship of interest. The node "U" represents the unmeasured confounders that can introduce residual confounding bias in the estimation.

**2.6.2. Exposure-response curve for currently observed temperature.** Causal exposure-response analysis is a method used to estimate the causal relationship between a continuous exposure (e.g., temperature) and an outcome (e.g., malaria incidence) while adjusting for confounding variables. Unlike traditional regression, which measures association, this approach isolates the causal effect by leveraging techniques like generalized propensity score or TMLE [43]. These methods account for observed confounders to simulate a hypothetical experiment where exposure levels are randomly assigned. The result is an exposure-response curve showing how changes in the exposure causally influence the outcome, even with observational data. Additionally, this curve identifies optimal exposure ranges and quantifies uncertainty via confidence intervals. We implemented this approach to the currently observed temperatures in the range of 15–30 °C (avoiding positivity assumption violation by extreme values), to model how temperature causally affects malaria incidence in the municipalities. We used the causal-curve Python package version 1.0.6 [43], applying the TMLE_Regressor method, with 2,500 trees, a max depth of four in each tree, and a learning rate of 0.001.

**2.6.3. Temperature regimes analysis.** The STR are causal inference methods that define interventions by probabilistically modifying treatment variables rather than imposing fixed rules. Unlike marginal structural models (e.g., marginal effect of setting all temperatures above 25 °C vs. setting all temperatures below 25 °C) or structural nested models (e.g., effect of temperature conditioned by humidity) [44], STR shift treatments according to a user-specified distribution [45,46]. For example, a modified treatment policy might additively increase each unit's observed temperature by 1 °C, but only if doing so stays within observed data support. This flexibility makes STR ideal for continuous treatments, where rigid interventions are unrealistic.

The STR can model incremental temperature rises while respecting natural variability. Suppose you want to estimate the effect of raising temperatures by 1–2 °C. An STR would shift each municipality's observed temperature by this range, weighted probabilistically, rather than assuming uniform exposure. The STR are particularly useful for our research question because they capture nuanced, population-level effects of climate variables. By defining a grid of counterfactual shifts (e.g., + 1 °C, + 2 °C), researchers can fit structural models to estimate exposure-response relationships between temperature and excess malaria cases (See S1 Text for more details).

We used the tmle3shift R package version 0.2.1 [47] to estimate the effect on excess malaria cases for the next regimes: 1) Observed temperature + 0.5 °C, 2) Observed temperature + 1.0 °C, 3) Observed temperature + 1.5 °C, and 4) Observed temperature + 2.0 °C. The tmle3shift package estimates the causal effect via TMLE, which adjusts for confounding while avoiding extrapolation beyond observed data. For instance, it ensures municipalities already at extreme temperatures aren't shifted beyond plausible limits, improving validity. TMLE's double robustness protects against model misspecification, and confidence intervals account for stochastic uncertainty. Note that we had to binarize the response variable and obtain the excess of cases for the STR model, due to the restriction of the package tmle3shift.

The Super Learner method is an ensemble machine learning approach that combines predictions from multiple algorithms to create a single model with optimal predictive performance. It operates by using cross-validation to evaluate and weight the performance of different algorithms, ultimately selecting the combination that minimizes prediction error [48]. This method is particularly useful for understanding causal effects because it allows for the flexible modeling of complex relationships in data while avoiding overfitting. By integrating diverse algorithms, the Super Learner can adapt to various data structures and estimate counterfactual outcomes, which are critical in causal inference studies. We used the sl3 R package version 1.4.3 [49] to implement the Super Learner method in estimating the effect on excess malaria cases for the temperature regimes defined in the STR model. The machine learning algorithms included in the Super Learner method were: Random forest, Bayesian additive regression trees, high-dimensional adaptive Lasso, and multivariate adaptive regression splines.

To facilitate model convergence, we transformed the values of continuous co-variables into binary values, using as threshold the median of each variable. The code and dataset for replicating the potential distribution of vectors, and the causal machine learning analysis are accessible at: https://github.com/juandavidgutier/temperature_malaria_STR.

## 3. Results

Between 2007 and 2023 were reported 1,114,580 malaria cases in Colombia, most of which occurred in males (797,500 cases). The year with more cases was 2010 with a total of 116,644 cases (10.5%). The age groups 10–19, 20–29, and 0–9 represented most of the cases with 25.8%, 23.8%, and 21.2% of the cases, respectively. The top three municipalities with more cases were El Bagre (103,300 cases), Caceres (59,248), and Puerto Libertador (54,754).

The DAG's analysis implemented with the R package ggdag showed that the correct adjustment had to include the variables: Year, month, atmospheric and oceanic indices, and rainfall, to obtain an unbiased estimate of the effect of temperature on malaria incidence. Note that the inclusion of the variables: Percentage of the area within each municipality suitable for supporting four or more vector species, and percentage of households with multidimensional poverty, will introduce collider bias, which occurs when two or more variables influence a common outcome (the collider). Adjustment on this collider, inadvertently introduces a spurious association between unrelated variables, leading to incorrect conclusions about the causal relationship of interest [28].

The estimation of the exposure-response curve for the effect of currently observed temperature on the SIR in the top 100 municipalities with the highest malaria cases showed a peak of the effect of temperature at 23.5 °C, where the SIR reached an average value of 50 (Fig 3), with a prominent rise between 20 and 23.5 °C. Values of temperature over 23.5 °C seem to reduce the SIR in these municipalities, and temperatures above 28.5 °C produced SIR values not different from 0, according to the 95% confidence interval (95% CI).

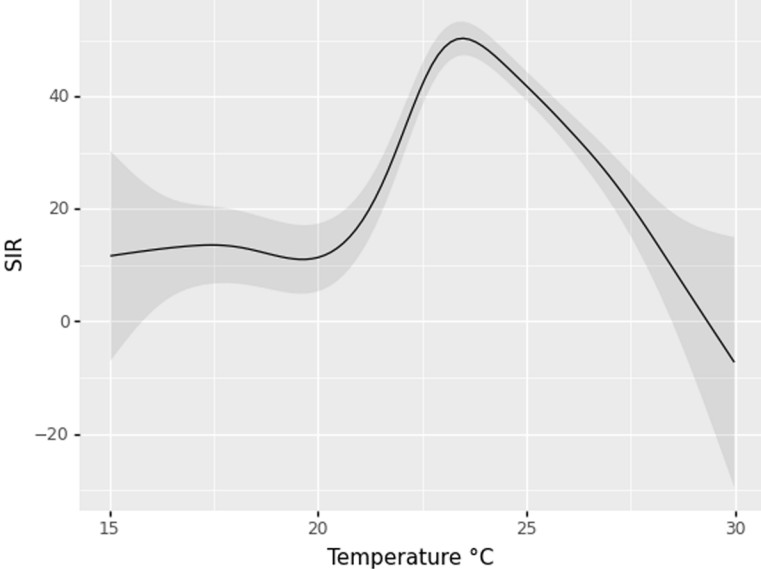

**Fig 3. Exposure-response curve with the effect of currently observed temperature on the Standardized Incidence Ratio (SIR) for the top 100 municipalities with the highest malaria cases in Colombia.** The black line represents the effect, and the gray band corresponds to the 95% CI.

The implementation of the STR method to estimate the Average Treatment Effect (ATE) on excess malaria cases for different temperature regimes revealed a progressive increase in the magnitude of the negative association as temperature increased in the top 100 municipalities with the highest malaria cases in Colombia (Fig 4). A 0.5 °C temperature increase was associated with a 0.7 percentage point reduction in the probability of excess malaria cases (ATE = -0.007, 95% CI: -0.015 − 0.001), compared with current temperature. Larger increments (1.0 °C, 1.5 °C, and 2.0 °C) showed progressively stronger inverse associations (ATE = -0.021, -0.040, -0.063, respectively) relative to current temperatures, with all confidence intervals excluding zero. This suggests an exposure-response relationship where higher temperature increases are associated with greater reductions in the probability of excess malaria cases (Fig 3).

## 4. Discussion

Our findings reveal a non-linear relationship between temperature and malaria incidence in high-burden Colombian municipalities. The exposure-response curve demonstrates that malaria transmission intensifies as temperatures increase from 15 °C to approximately 23.5 °C, after which the effect decreases at temperatures exceeding 23.5 °C. This non-linear pattern aligns with previous research by Mordecai et al., who identified an optimal temperature window for malaria transmission between 25–27 °C, with reduced transmission efficiency at both lower and higher temperatures [50]. The observed decline in the SIR at temperatures above 23.5 °C supports the biological understanding that extreme heat can impair mosquito survival and parasite development, despite accelerating other aspects of transmission [51].

The STR analysis provides critical insights into potential climate change scenarios. Our finding that the ATE progressively diminishes with progressive temperature increases, contradicts some predictions that climate warming will universally intensify malaria transmission [52]. Instead, our results suggest that in Colombia's highest-incidence municipalities—many of which already experience temperatures near or above the optimal range for malaria transmission—further warming might actually constrain transmission intensity. These findings align with previous works demonstrating that climate change may shift rather than uniformly expand malaria's geographical distribution [53]. This

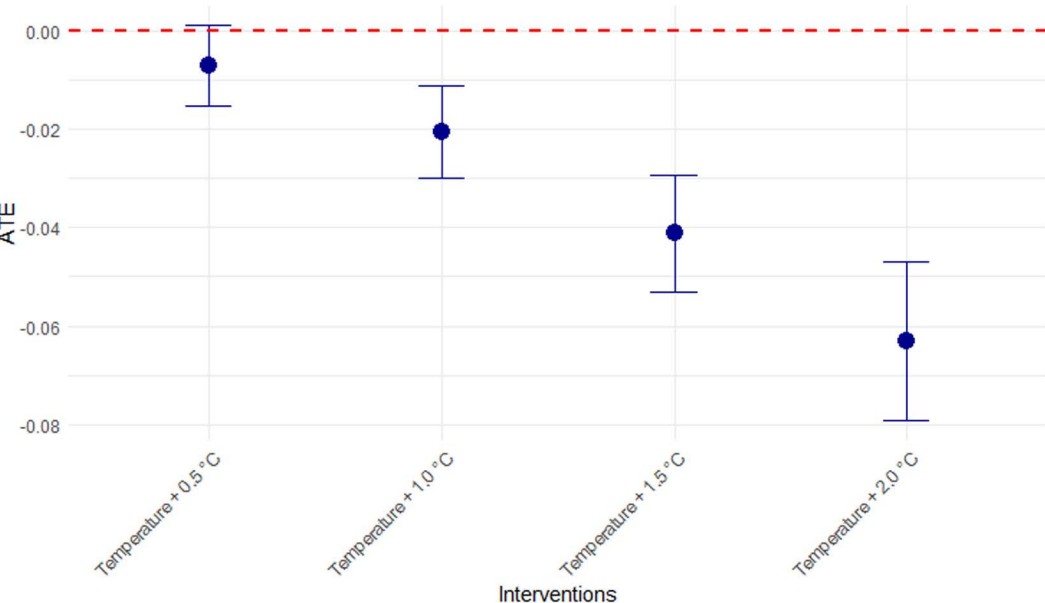

**Fig 4. Effect of the different temperature regimes evaluated on excess malaria cases.** The y-axis represents the Average Treatment Effect (ATE). The blue points depict the point estimates of the ATE for each temperature regime, compared with the current temperature. The vertical blue line extending from each blue circle represents the upper and lower bounds of the corresponding 95% CI. The dotted red line indicates the null effect (ATE = 0).

nuanced understanding is essential for accurate climate-health risk assessments in tropical regions like Colombia, where many endemic areas already experience mean temperatures exceeding 25 °C.

The causal machine learning approach employed in this study addresses the limitations of traditional statistical methods in climate-health research. By implementing a causal machine learning approach, we were able to disentangle the causal effects of temperature from confounding factors. This methodological strategy represents an improvement over correlation/regression-based approaches that have dominated previous research [6,36,54].

The spatial heterogeneity in malaria transmission dynamics across Colombia merits special attention in interpreting our results. The declining effect of temperature increases in high-incidence municipalities likely reflects the adaptation of local vector populations to specific thermal niches. *An. darlingi* and *An. albimanus*, the primary vectors in many Colombian endemic regions, have demonstrated varying temperature sensitivities and behavioral adaptations that influence their vectorial capacity [55,56].

The dependence on reported malaria cases introduces potential information biases, as records may be affected by underreporting, misdiagnosis, or differences in healthcare access. Although SIVIGILA provides comprehensive health data from medical professionals nationwide, it suffers from challenges like underreporting and varying data quality, particularly between urban and rural areas [57]. Additionally, the system's dependence on health facility reporting may overlook asymptomatic cases or individuals who do not seek/find medical care, leading to an underestimation of disease incidence. Our efforts to tackle information bias cannot fully address all sources of information bias inherent in an ecological study based on secondary data. To enhance the study's reliability, it is essential to obtain more granular data. This would help refine the understanding of how temperature influences malaria incidence, addressing the limitations posed by the current data collection methods.

A limitation of our study concerns the ecological design and its susceptibility to ecological fallacy [58]. Our municipality-level analysis examines the relationship between temperature and malaria incidence at a geographical unit

that may not accurately reflect individual-level exposure-outcome relationships [59]. The ecological fallacy presents particular challenges in climate-health research, as environmental exposures measured at municipal scales may not correspond to actual individual exposures due to micro-environmental variations, population mobility patterns, and differential access to protective measures such as bed nets or indoor residual spraying. Furthermore, the heterogeneity in socio-economic conditions, healthcare access, and behavioral factors within municipalities cannot be adequately captured in our aggregate analysis, which may lead to the misestimation of effect sizes [60]. While ecological studies remain valuable for identifying population-level patterns and informing public health policy, the inability to establish individual-level causality represents a significant constraint in interpreting our findings for targeted interventions [61]. Future research should incorporate individual-level data or multilevel modeling approaches to validate our ecological findings and minimize the risk of ecological fallacy in climate-malaria research.

Causal exposure-response is a method particularly useful for policy planning, as it identifies temperature thresholds where malaria risk escalates, guiding climate adaptation strategies. However, as any causal inference method, it relies on strict assumptions: 1) ignorability (all confounders are measured), 2) positivity (all municipalities have non-zero probability of experiencing any temperature level), and 3) consistency (there are not multiple versions of the treatment). Violations—like unmeasured confounders or sparse data at extreme temperatures (i.e., positivity assumption)—can bias results [43].

Similarly, the positivity assumption is a challenge in STR models, which require sufficient data support across the entire range of counterfactual temperature scenarios [12]. In our analysis of the top 100 Colombian municipalities with more malaria cases, we restricted our analysis to the range of 15 to 30 °C, avoiding positivity assumption violation by extreme values. However, the positivity assumption may have been violated at extreme temperature values, particularly when applying the +2.0 °C shift to municipalities already experiencing temperatures at the upper boundary of our observed data. Previous analyses have noted that positivity violations in environmental epidemiology can lead to biased effect estimates and artificially narrow confidence intervals, potentially overstating the precision of our findings [62,63].

Our research may suffer from selection bias by limiting the analysis (by computational restrictions) to the municipalities with the highest malaria incidence, which could compromise the generalizability of our findings to other municipalities or regions with different epidemiological profiles [64]. By focusing exclusively on high-incidence municipalities, our study may have excluded areas with different climate-malaria relationships, such as municipalities at the threshold of malaria transmission zones or those with emerging transmission patterns due to environmental changes [50]. This geographic selection could result in an overrepresentation of municipalities with specific environmental, socio-economic, or vector ecology characteristics that facilitate high malaria transmission, potentially limiting the external validity of our temperature-malaria exposure-response curves. Future research should include a broader sampling strategy that encompasses municipalities across the full spectrum of malaria transmission to enhance the generalizability of climate-malaria relationships and improve the external validity of causal machine learning applications in tropical disease epidemiology.

While multidimensional poverty represents a collider variable in our causal framework (and for this reason was excluded from the estimation, to avoid collider bias), understanding how socioeconomic conditions modify the temperature-malaria relationship remains of substantial epidemiological interest. Poverty is historically associated with malaria since families in areas with lower economic status usually have less access to quality housing, health services, and sanitation services [65]. In the context of temperature-malaria relationships, multidimensional poverty may modify the causal effect by influencing factors that could amplify or attenuate the impact of thermal conditions on transmission dynamics [4]. Future research should explore other causal frameworks, while carefully avoiding conditioning on the collider variable itself. Such investigations could inform targeted interventions that address both climatic and socioeconomic determinants of malaria transmission, particularly relevant for Colombia's vulnerable Indigenous and Afro-descendant populations, who experience disproportionate poverty burdens in malaria-endemic regions [4].

The double robustness property of TMLE, while theoretically protective against model misspecification, still relies on either the outcome model or the propensity score model being correctly specified [66]. In complex ecological systems

like malaria transmission, where temperature effects may operate with various interaction ways, the specification of these models becomes particularly challenging. In our implementation, we utilized an algorithm ensemble to mitigate this concern, but the binary transformation of continuous co-variables to facilitate convergence may have oversimplified the intricate relationships present in the data.

A limitation of our STR approach concerns temporal considerations in malaria epidemiology. The tmle3shift package implements point-in-time shifts in temperature without inherently accounting for the lag structures between temperature changes and subsequent malaria outcomes. Malaria transmission dynamics involve multiple temperature-dependent processes, including mosquito development, parasite incubation, and human immune responses. Our monthly aggregation of data may have obscured these temporal dynamics, potentially masking critical non-linear relationships between temperature fluctuations and disease incidence. Previous approaches have worked on distributed lag non-linear models suggesting that the failure to account for these lagged effects can bias estimates of climate-health relationships, particularly when delayed impacts are present [67].

The computational implementation of STR within tmle3shift presents additional challenges that may influence the reliability of our results. Convergence issues in TMLE estimation, particularly when incorporating complex machine learning algorithms in the Super Learner ensemble, can lead to unstable estimates or excessive variance [68]. In our implementation, we addressed convergence challenges by binarizing continuous co-variables, a pragmatic solution that nonetheless sacrifices information and may introduce threshold effects not present in the underlying data. Furthermore, the tmle3shift package currently lacks built-in functionality for incorporating clustered or spatially correlated data structures, which are inherent in a municipality-level analysis [69]. This limitation may have resulted in underestimated standard errors and overly narrow confidence intervals, as the spatial autocorrelation between neighboring municipalities was not formally modeled.

Finally, the generalizability of our STR findings is constrained by the theoretical framework underpinning the counterfactual scenarios. The uniform temperature shifts applied across municipalities inherently assume that climate change scenarios will manifest as consistent warming across Colombia's diverse geographical regions. However, climate models project heterogeneous patterns of warming with regional variation [70]. The tmle3shift implementation lacks mechanisms to incorporate this spatial heterogeneity in projected temperature changes, potentially limiting the real-world applicability of our findings. Additionally, the package's current framework cannot directly incorporate uncertainty in the magnitude of future temperature increases, which climate scientists have identified as substantial [71]. Future methodological developments should aim to integrate ensemble climate projections with STR methods to better characterize the range of plausible temperature effects on malaria under various climate change scenarios.

## 5. Conclusion

Through causal machine learning methods, our study provides insights into the causal relationship between temperature and malaria incidence in high-burden Colombian municipalities. Our findings reveal a non-linear temperature-malaria relationship with transmission intensity peaking at 23.5 °C and subsequently declining at higher temperatures. This pattern challenges simplistic assumptions about climate change uniformly increasing malaria risk and suggests that in many high-incidence municipalities of Colombia—where temperatures already approach or exceed optimal transmission thresholds—further warming may actually constrain malaria transmission intensity rather than amplify it. The progressive reduction in the ATE provides quantitative evidence for this counterintuitive relationship.

The methodological framework established in this research demonstrates the value of causal inference approaches for disentangling complex climate-health relationships. By explicitly modeling the effect of temperature on malaria incidence through causal machine learning, we move beyond correlation/association-based analyses that have dominated previous climate-malaria research. Despite several methodological limitations—including information bias, challenges in modeling temporal dynamics, and the assumption of uniform temperature shifts—our analytical approach represents an advancement in understanding how climate change can causally influence infectious disease transmission. This framework can

be adapted to study other vector-borne diseases and climate-sensitive health outcomes across diverse geographical contexts.

Our results have implications for malaria control strategies and climate change adaptation planning in Colombia. Public health authorities should adopt municipality-specific approaches that consider the local temperature profiles and their position relative to the optimal transmission window. In particular, municipalities currently experiencing average temperatures below 23.5 °C may face increased transmission risk with warming and should strengthen surveillance and vector control programs accordingly. Conversely, in municipalities where temperatures already exceed 23.5 °C, resources might be better directed toward addressing other determinants of malaria transmission, such as access to healthcare, housing improvements, and targeted interventions for vulnerable populations. Future research should integrate lagged effects and spatially heterogeneous climate projections to refine our understanding of how changing temperature regimes will reshape Colombia's malaria landscape in the coming decades.

## Supporting information

**S1 Text. Temperature regimes analysis.**
(DOCX)

## Acknowledgments

We thank the Colombian Ministry of Health for providing access to epidemiological data. We also express our gratitude to Mariano Altamiranda and Julián Ávila for their assistance with the methods for assessing the potential occurrence of malaria vectors.

## Author contributions

**Conceptualization:** Juan David Gutiérrez.

**Data curation:** Juan David Gutiérrez.

**Formal analysis:** Juan David Gutiérrez.

**Funding acquisition:** Juan David Gutiérrez.

**Investigation:** Juan David Gutiérrez.

**Methodology:** Juan David Gutiérrez.

**Project administration:** Juan David Gutiérrez.

**Resources:** Juan David Gutiérrez.

**Software:** Juan David Gutiérrez.

**Supervision:** Juan David Gutiérrez.

**Validation:** Juan David Gutiérrez.

**Visualization:** Juan David Gutiérrez.

**Writing – original draft:** Juan David Gutiérrez.

**Writing – review & editing:** Juan David Gutiérrez.

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
