## [Decision Letter · Decision Letter 0]

18 Jul 2025

PGPH-D-25-00666

Stochastic Treatment Regimes in Climate-Health Research: Reassessing Malaria Risk Under Warming Scenarios in Colombia

Dear Dr. Gutiérrez,

Thank you for submitting your manuscript to PLOS Global Public Health. After careful consideration, we feel that it has merit but does not fully meet PLOS Global Public Health’s publication criteria as it currently stands. Therefore, we invite you to submit a revised version of the manuscript that addresses the points raised during the review process.

Please note that we have only been able to secure a single reviewer to assess your manuscript. We are issuing a decision on your manuscript at this point to prevent further delays in the evaluation of your manuscript. Please be aware that the editor who handles your revised manuscript might find it necessary to invite additional reviewers to assess this work once the revised manuscript is submitted. However, we will aim to proceed on the basis of this single review if possible. Could you please carefully revise the manuscript to address all comments raised?

We look forward to receiving your revised manuscript.

Kind regards,

Helen Howard

Staff Editor

Journal Requirements:

1. We have amended your Competing Interest statement to comply with journal style. We kindly ask that you double check the statement and let us know if anything is incorrect.

Reviewers' comments:

Reviewer's Responses to Questions

**Comments to the Author**

1. Does this manuscript meet PLOS Global Public Health’s publication criteria? Is the manuscript technically sound, and do the data support the conclusions? The manuscript must describe methodologically and ethically rigorous research with conclusions that are appropriately drawn based on the data presented.

Reviewer #1: Yes

2. Has the statistical analysis been performed appropriately and rigorously?

Reviewer #1: I don't know

3. Have the authors made all data underlying the findings in their manuscript fully available (please refer to the Data Availability Statement at the start of the manuscript PDF file)?

Reviewer #1: Yes

4. Is the manuscript presented in an intelligible fashion and written in standard English?

Reviewer #1: Yes

5. Review Comments to the Author

Reviewer #1: I see this paper more of a confirmation of existing studies on the effects of temperature on causing malaria. In addition to the temperature as the input variables, the approaches consider 11 other atmospheric and oceanic indices acting as confounding factors.

The study does not address how its aggregate findings might limit conclusions about individual-level malaria risk, which is a fundamental limitation of the ecological study design in terms of its scientific approach and the scope of its inferences. There are other studies considering socio-economic factors and while the authors refer to some effectors of colonization, they were not considered as a variable. An association found between temperature and malaria incidence at the municipality level might not necessarily hold true at the individual/household level, or that the observed patterns for municipalities do not directly translate to individual risk factors. The study does not address how its aggregate findings might limit conclusions about individual-level malaria risk

The authors do not discuss the potential for selection bias inherent in studying only a subset of municipalities based on their outcome (high malaria incidence). The causal relationship between temperature and malaria transmission, including the optimal temperature range and the Average Treatment Effect (ATE), might differ in municipalities with lower malaria incidence or different epidemiological profiles. Therefore, the findings regarding temperature effects and future warming scenarios, while valuable for the studied areas, may not be broadly generalizable or valid for areas with different baseline malaria burdens or environmental characteristics within Colombia or elsewhere. This constitutes a limitation in the external validation and broader applicability of the scientific findings.

The methodology used (e.g., DAG or causal machine learning) has not been cited and I believe any references to previous work that inspires this work will be appreciated.

6. PLOS authors have the option to publish the peer review history of their article (what does this mean?). If published, this will include your full peer review and any attached files.

**Do you want your identity to be public for this peer review?** For information about this choice, including consent withdrawal, please see our Privacy Policy.

Reviewer #1: **Yes: **Dr. Hamid Mukhtar

---

## [Editor Report · Decision Letter 1]

7 Sep 2025

PGPH-D-25-00666R1

Stochastic Treatment Regimes in Climate-Health Research: Reassessing Malaria Risk Under Warming Scenarios in Colombia

Dear Dr. Gutiérrez,

Thank you for submitting your manuscript to PLOS Global Public Health. After careful consideration, we feel that it has merit but does not fully meet PLOS Global Public Health’s publication criteria as it currently stands. Therefore, we invite you to submit a revised version of the manuscript that addresses the points raised during the review process.

Specifically, further explanation on the model and variables are required. In addition, since there are many analysis steps and methods involved, it would be good to provide an overview diagram for the readers to understand how these fit together.

We look forward to receiving your revised manuscript.

Kind regards,

Michele Nguyen

Academic Editor

Journal Requirements:

Additional Editor Comments:

Line 17, Abstract: Please spell out "approximately" in full.Line 91: It would be good to add a flowchart of the analysis steps and methods used at the start of the Methods section, highlighting the purpose of each step. Line 141: Please provide details on the 19 bioclimatic variables and the final model formula for the vectors' co-occurrence model. How was variable selection conducted? Line 186-187: Why did you choose to consider the percentage of area within each municipality suitable for supporting four or more vector species? E.g. why not two or more?Line 191-195: Please provide references to support the statement that successive governments have systematically overlooked in terms of social advancement and provision of essential services for communities inhabited mainly by Indigenous and Afro-descendant populations, as a result of Spanish colonizers identifying these areas as having unhealthy climates. Line 259: In addition to the code for the causal machine learning analysis, it would be good to provide the code for the vectors' co-occurrence model.

---

## [Editor Report · Decision Letter 2]

12 Sep 2025

Stochastic Treatment Regimes in Climate-Health Research: Reassessing Malaria Risk Under Warming Scenarios in Colombia

PGPH-D-25-00666R2

Dear Mr. Gutiérrez,

We are pleased to inform you that your manuscript 'Stochastic Treatment Regimes in Climate-Health Research: Reassessing Malaria Risk Under Warming Scenarios in Colombia' has been provisionally accepted for publication in PLOS Global Public Health.

Best regards,

Michele Nguyen

Academic Editor